# *Francisella tularensis* Glyceraldehyde-3-Phosphate Dehydrogenase Is Relocalized during Intracellular Infection and Reveals Effect on Cytokine Gene Expression and Signaling

**DOI:** 10.3390/cells12040607

**Published:** 2023-02-13

**Authors:** Ivona Pavkova, Monika Kopeckova, Marek Link, Erik Vlcak, Vlada Filimonenko, Lenka Lecova, Jitka Zakova, Pavlina Laskova, Valeria Sheshko, Miloslav Machacek, Jiri Stulik

**Affiliations:** 1Department of Molecular Pathology and Biology, Faculty of Military Health Sciences, University of Defence, Trebesska 1575, 500 01 Hradec Kralove, Czech Republic; 2Institute of Molecular Genetics of the Czech Academy of Sciences, Electron Microscopy Core Facility, Videnska 1083, 142 20 Prague, Czech Republic; 3Institute of Molecular Genetics of the Czech Academy of Sciences, Department of Biology of the Cell Nucleus, Videnska 1083, 142 20 Prague, Czech Republic; 4Department of Biochemical Sciences, Faculty of Pharmacy in Hradec Kralove, Charles University in Prague, Akademika Heyrovskeho 1203, 500 05 Hradec Kralove, Czech Republic

**Keywords:** multitasking, pleiotropy, *Francisella*, glyceraldehyde-3-phosphate dehydrogenase, infection, secretion, interacting partners

## Abstract

Glyceraldehyde-3-phosphate dehydrogenase (GAPDH) is known for its multifunctionality in several pathogenic bacteria. Our previously reported data suggest that the GAPDH homologue of *Francisella tularensis*, GapA, might also be involved in other processes beyond metabolism. In the present study, we explored GapA’s potential implication in pathogenic processes at the host cell level. Using immunoelectron microscopy, we demonstrated the localization of this bacterial protein inside infected macrophages and its peripheral distribution in bacterial cells increasing with infection time. A quantitative proteomic approach based on stable isotope labeling of amino acids in cell culture (SILAC) combined with pull-down assay enabled the identification of several of GapA’s potential interacting partners within the host cell proteome. Two of these partners were further confirmed by alternative methods. We also investigated the impact of *gapA* deletion on the transcription of selected cytokine genes and the activation of the main signaling pathways. Our results show that ∆*gapA*-induced transcription of genes encoding several cytokines whose expressions were not affected in cells infected with a fully virulent wild-type strain. That might be caused, at least in part, by the detected differences in ERK/MAPK signaling activation. The experimental observations together demonstrate that the *F. tularensis* GAPDH homologue is directly implicated in multiple host cellular processes and, thereby, that it participates in several molecular mechanisms of pathogenesis.

## 1. Introduction

Glyceraldehyde-3-phosphate dehydrogenase (GAPDH) was one of the first multitasking proteins to be defined. Its multifunctionality has been demonstrated across all domains of life. In addition to glycolysis, GAPDH has been shown to be involved in a number of other cellular nonmetabolic functions (reviewed in [1,2]) that are usually linked with its distinct cellular localization and/or posttranslational modification [3,4]. In various pathogenic bacteria, the atypically localized GAPDH has been shown to play a role in processes associated with their virulence and pathogenesis [5]. For example, surface-exposed homologues of *Streptococcus* spp., *Staphylococcus aureus*, *Mycoplasma* spp., and pathogenic *E. coli* exhibit various adhesive functions, thus facilitating the colonization and invasion of the host tissues [6,7]. *Mycobacterium tuberculosis* utilizes GAPDH as a receptor for lactoferrin and transferrin to acquire iron from host resources, which is essential for its survival and replication inside the host cell [8,9]. The immunomodulatory activities of extracytosolic bacterial GAPDH have also been described [10,11,12]. In *Streptococcus pyogenes*, it is able to capture the C5a component of the complement system; in *Streptococcus agalactiae*, it exerts a stimulatory effect on lymphocytes and interleukin-10 production. To date, *Listeria monocytogenes* is the only intracellular bacterium that has been demonstrated to use GAPDH for manipulating phagosome maturation by targeting the Rab proteins function to facilitate its vacuolar escape to the host cell cytosol [13].

*Francisella tularensis* is a facultative intracellular pathogenic bacterium that is a causative agent of the zoonotic disease tularemia. Two *F. tularensis* subspecies—*tularensis* (type A) and *holarctica* (type B)—are associated with human disease. Due to its high infectivity, its easy spread by aerosol, its ability to cause acute respiratory infections with high mortality rates, and its lack of a licensed vaccine, *F. tularensis* is considered a potential bioterrorism agent and is classified by the Centers for Disease Control as a category A agent [14,15,16]. *F. tularensis* primarily targets macrophages [17] and its intramacrophage cycle involves cell entry by so-called “looping phagocytosis” [18] and inhibition of phagosome–lysosome fusion, followed by a rapid escape from the phagosome to the host cytosol, where it replicates to high numbers [19]. As an intracellular pathogen, it is able perfectly to evade recognition and destruction by the host immune system. It employs multiple survival strategies for this purpose, including intracellular replication, the atypical structure of lipopolysaccharide, and the aberrant activation of the immune response [20].

Despite intensive research, knowledge of the precise molecular mechanisms of the pathogeneses of *F. tularensis* remains incomplete. Moreover, there is a quite large discrepancy within the current knowledge, as many studies were performed on nonpathogenic *F. novicida* or an attenuated live-vaccine strain (LVS), which obviously differ in their capabilities to evade the host immune system compared to the fully pathogenic strains [20,21,22]. To date, several determinants essential for *F. tularensis* intracellular survival have been determined. They might be divided into several groups. One group includes genes encoding proteins required for the phagosomal escape and involves, for example, genes of the *Francisella* pathogenicity island (e.g., *iglC*, *iglI*, *iglD*, *vgrG*, and *mglA*), hypothetical lipoproteins encoded by loci FTT1103 and FTT1676, or several genes of the pyrimidine biosynthesis pathway. Moreover, a number of factors required for cytosolic replication could also be identified, such as genes involved in purine metabolism or a number of other genes encoding proteins with unknown functions (e.g., FTT0369c and FTT0989) [20]. The deletion of several genes (e.g., *tolC*, *ripA*, and *fopA*) in other mutant strains resulted in higher rates of cytotoxicity. In this way, the bacterium loses its intracellular replicative niche, which consequently decreases proliferation. These so-called hypercytotoxic mutants are not able to dampen the cell protective mechanisms effectively, as they display increased lysis accompanied by the release of pathogen-associated molecular patterns into the host cell milieu, resulting in inflammasome activation and an increased proinflammatory response [23,24].

In a recently published study, we presented a mutant of the fully virulent *F. tularensis* subsp. *holarctica* FSC200 strain with a deleted gene *gapA* encoding GAPDH homologue (GapA) [25]. We demonstrated that the GapA protein contributes to fully virulent manifestation in both in vitro and in vivo models of infection. The *gapA* mutant strain revealed decreased proliferation inside macrophages and significantly reduced virulence in mice and was even able to induce protection in mice challenged with a parental wild-type strain. Furthermore, the extracellular localization of GapA in bacteria cultivated in medium was shown, indicating the multifunctionality of this protein. In our next study, we were able to identify several bacterial proteins as interaction partners of GapA, thus suggesting its implication in DNA-repair processes and in the subcellular distribution of some proteins [26]. The main question concerning the potential role of GapA in pathogenesis remains open, however, and was the main impulse for the study presented here.

First, we analyzed the localization of GapA in bacteria proliferating inside primary macrophages and confirmed its surface exposition and occurrence inside the host cell milieu using transmission electron microscopy (TEM). Next, we employed a high-throughput screen based on the stable isotope labeling of amino acids in cell culture (SILAC) in combination with pull-down assay and quantitative mass spectrometry analysis (LC-MS/MS) for the detection and identification of bacterial GapA’s potential host interaction partners. Two selected hits were successfully verified by a distinct methodological approach, and one of these hits, the S100A6, was subjected for preliminary functional characterization. We additionally analyzed the impact of *gapA* gene deletion on cytokine production and host signaling processes to acquire deeper insights into the molecular mechanisms that are affected differently by this attenuated mutant strain compared to the fully virulent wild-type strain.

## 2. Materials and Methods

### 2.1. Francisella Strains, Growth Conditions

*Francisella tularensis* subsp. *holarctica* strain FSC200 (wt) from the *Francisella* strain collection was kindly provided by Åke Forsberg, Swedish Defense Research Agency, Umea, Sweden [27]. The derived mutant strain with a deleted *gapA* gene (∆FTS_1117/FSC200, ∆*gapA*) had been prepared and characterized previously [25]. All the *F. tularensis* strains were cultured on McLeod agar enriched with bovine hemoglobin (Becton Dickinson, Cockeysville, MD, USA) and IsoVitaleX (Becton Dickinson) at 37 °C.

### 2.2. Bone Marrow-Derived Macrophages (BMDM) Cultivation and Infection

BMDMs were generated from bone marrow cells collected from the femurs and tibias of 6- to 10-week-old female BALB/c mice by differentiation for 7 days in Dulbecco’s Modified Eagle’s Medium (DMEM, Sigma–Aldrich, St. Louis, MO, USA) supplemented with 10% fetal bovine serum and 20% L929-conditioned medium (as a source of macrophage colony-stimulating factor [M-CSF]) [28]. The differentiated BMDMs were seeded on cultivation plates and allowed to adhere overnight. Cells were infected with *F. tularensis* strains at a multiplicity of infection (MOI) of 50. To synchronize the infection, plates were centrifuged at 400× *g* for 3 min (t = 0). In experiments with infection times longer than 1 h, extracellular bacteria were killed by gentamicin treatment (5 µg/mL) for 30 min. The cells were kept at 37 °C in 5% CO_2_ for the indicated time post-infection (p.i.).

### 2.3. Quantitative Real-Time Polymerase Chain Reaction

RNA was isolated from BMDMs infected with wt or ∆*gapA* for 8 h using RNeasy kit from Qiagen (Hilden, Germany). One microgram of total RNA was reverse-transcribed using oligo (dT) primers (New England Biolabs, Ipswich, MA, USA). Quantitative real-time PCR analysis was performed and analyzed using the ABI Prism 7500 Fast RT-PCR System (Applied Biosystems, Waltham, MA, USA). The results were normalized to the housekeeping gene 18S rRNA (Rn18S1) and expressed as fold change relative to RNA samples from mock-treated cells using the comparative Ct method (ΔΔCt). The following TaqMan Gene Expression Assays (Applied Biosystems) were used: *Il1b* (Mm00434228_m1), *Tnf* (Mm00443258_m1), *Il10* (Mm01288386_m1), *Il12b* (Mm01288989_m1), *Ifnb1* (Mm00439552_s1), *Il6* (Mm00446190_m1), *Nos2* (Mm00440502_m1), *Arg1* (Mm00475988_m1), and *housekeeping* gene *Rn18s* (Mm03928990_g1).

### 2.4. Immunoblot Analysis

Cells were lysed at indicated times p.i. in RIPA buffer supplemented with EDTA-free complete protease inhibitor mixture (Roche, Basel, Switzerland) and phosphatase inhibitors (Cocktail set II, Merck, Darmstadt, Germany). Total protein concentration was determined using a bicinchoninic acid assay. Protein samples were separated on SDS-PAGE and transferred to polyvinylidene difluoride membranes (BioRad, Hercules, CA, USA). Blots were blocked with 5% milk and incubated with different primary antibodies overnight, followed by incubation with a secondary antibody conjugated with horse-radish peroxidase (Agilent Dako, Santa Clara, CA, USA). Proteins were detected using a BM Chemiluminescence Blotting Substrate (POD) (Roche) or SuperSignal™ West Femto Maximum Sensitivity Substrate (Thermo Fisher Scientific, Waltham, MA, USA) on X-ray films.

Primary antibodies anti-p44/42 MAPK (Erk1/2), anti-phospho-p44/42 MAPK (Erk 1/2), anti-p38MAPK, anti-phospho-p-38 MAPK, anti-JNK2, anti-phospho-SAPK/JNK, anti-IκBα, and anti-phospho-IκBα were purchased from Cell Signaling Technology (Danvers, MA, USA); anti-IL-1β and anti-phospho-IL-1β were purchased from Abcam (Cambridge, UK); anti-Actin and anti-GAPDH for loading controls were purchased from Sigma-Aldrich.

### 2.5. Transmission Electron Microscopy

BMDMs seeded on glass coverslips at density 1 × 10^6^ cells per well in 24-well tissue culture were infected with *F. tularensis* FSC200, as described above, for 2, 12, and 24 h. Following infection, the cells were briefly washed in pre-warmed Sörensen’s buffer (0.1 M Na/K phosphate buffer, pH 7.2–7.4) and fixed in 2.5% paraformaldehyde with 0.25% glutaraldehyde (Thermo Fisher Scientific) in Sörensen’s buffer for 1 h at room temperature (RT). All subsequent steps were performed on ice. After several washes, free aldehyde groups were quenched with 0.02 M glycine in Sörensen’s buffer for 10 min. Samples were then dehydrated in a series of ethanol (Lach-Ner, Neratovice, Czech Republic) and embedded into LR White resin (Sigma-Aldrich). After polymerization for 72 h under UV light at 4 °C, 80 nm ultrathin sections were prepared using the Ultramicrotome Leica EM UC6 (Leica Microsystems, Wetzlar, Germany) equipped with a diamond knife (Diatome, Biel, Switzerland). The sections were mounted on formvar-coated 3.05 mm gilded copper slots (Agar scientific, Essex, UK), immunogold-labeled following a conventional protocol [29], and examined in an FEI Morgagni 268 transmission electron microscope (TEM) with Mega View III CCD camera (Olympus Soft Imaging Solutions, Münster, Germany) or in a Jeol JEM-1400 FLASH TEM equipped with 2kx2k Matataki CMOS camera. Both TEMs were operated at 80 kV. Antibodies used for immunolabeling were primary rabbit anti-FTT1368 (GapA) antibody at dilutions 1:25 and 1:100 (Apronex, Vestec, Czech Republic) and secondary goat anti-rabbit IgG (H + L) antibody coupled with 12 nm colloidal gold particles (Jackson ImmunoResearch Laboratories Inc., Baltimore Pike, West Grove, PA, USA; 111-205-144; dilution 1:40). Uninfected BMDMs were used as a control of specificity of the GapA antibody, and usual technical negative controls were performed with an omitted primary antibody. The density of immunolabeling in specified compartments of bacterial cells and host cells was calculated as a ratio between the number of gold nanoparticles in a specified region and its area in µm^2^. The areas were measured in ELLIPSE Software, version 2.0.8.1 (ViDiTo, Kosice, Slovakia). The calculation of statistical significance was based on a comparison of labeling density values and variance in individual cells between analyzed variants.

### 2.6. Immunofluorescence

BMDMs attached to glass coverslips (at density 1 *×* 10^5^ in 24-well plate) were infected with wt or ∆*gapA*, as described above. Cells were washed 1 h post-infection with phosphate-buffered saline (PBS), fixed with 4% paraformaldehyde (pH 7.5) for 20 min, washed with PBS again, then blocked and permeabilized with 5% normal goat serum and 0.3% Triton X-100 in PBS for 60 min at RT. The cells were stained for endogenous NF-κB p65 using rabbit anti-NF-κB p65 (Abcam, Cambridge, UK) (1:100) in 1% bovine serum albumin, 0.3% Triton X-100 in PBS overnight at 4 °C, followed by Alexa 546-conjugated goat anti-rabbit IgG (1:500) for 1 h at RT. The cells were washed three times with PBS and stained with Hoechst 33,342 (Thermo Scientific, Waltham, MA, USA) for 20 min at RT to visualize the nuclei, mounted on glass slides with ProLong Gold Antifade Mountant (Invitrogen, MA, USA) and imaged on Nikon Ti-E epifluorescence microscope and A1+ laser scanning confocal microscope (Nikon, Tokyo, Japan). The image processing was performed using NIS Elements AR 4.2 (Laboratory Imaging, Prague, Czech Republic).

### 2.7. GapA Protein Expression and Purification

For the expression and purification of recombinant GapA protein tagged with Twin-Strep, the previously prepared construct of *gapA* C-terminally fused with one Strep-tag encoding sequence (*gapA*-Strep) was used [26]. The second Strep-tag sequence was added using PCR amplification with specific primers (Forward: 5′-GCCCTCGAGTTATTTCTCGAACTGCGGGTGGCTCC-3′, Reverse: 5′-GCGCCATGGTGAGAGTTGCAATTAATGGTTTCGGTAGAATTGGT-3′). The final construct *gapA*-Twin-Strep (WSHPQFEK-GGGSGGGSGGS-SAWSHPQFEK) was inserted into a pET28b expression vector between *NcoI* and *XhoI* restriction sites. Expression vector pET-*gapA*-Twin-Strep was transformed into *E. coli* NiCo21(DE3) (NEB, Ipswich, MA, USA) and the cells were grown in Express™ Instant TB Medium (Novagen, Merck KGaA, Darmstadt, Germany) overnight at 28 °C. Pelleted bacteria were resuspended in buffer containing 100 mM Tris/HCl, pH 7, 150 mM NaCl (pH 7.6), benzonase (150 U/mL, Merck), and an EDTA-free complete protease inhibitor mixture (Roche), then lysed by two passages through a French pressure cell at 16,000 psi. The lysate was then incubated with an appropriate amount of Avidin (11 U/mg; IBA Lifesciences, Göttingen, Germany) for 15 min at 4 °C and clarified by centrifugation at 11,000× *g* at 4 °C. The clarified lysate was incubated with magnetic beads MagStrep “type3” XT (IBA-Lifesciences) for 30 min on ice for purification of Twin-Strep-tagged GapA protein. The beads were washed three times with wash buffer (100 mM Tris/HCl pH 8.0, 150 mM NaCl, 1 mM EDTA) and bound protein was eluted with elution buffer (100 mM Tris/HCl, pH 8.0, 150 mM NaCl, 1 mM EDTA, 50 mM biotin). Finally, the elution buffer was changed to 100 mM Tris/HCl, pH 8.0, 150 mM NaCl buffer. Eluted GapA protein was verified by SDS-PAGE, followed by Coomassie staining, and the protein concentration was determined using a bicinchoninic acid assay.

### 2.8. SILAC-Pull-Down in J.774.2

A murine monocyte-macrophage cell line J774.2 (ATCC) was grown in DMEM supplemented with 10% fetal bovine serum. For metabolic labeling, the cells were transferred into arginine- and lysine-free DMEM (DMEM for SILAC, Thermo Fisher Scientific) supplemented with 10% dialyzed fetal bovine serum (Invitrogen) and heavy labeled amino acids L-arginine hydrochloride [^13^C_6_ ^15^N_4_] and L-lysine hydrochloride [^13^C_6_ ^15^N_2_] (Sigma-Aldrich) in the same concentrations as in the standard DMEM medium (“heavy” medium). After five cell divisions, the incorporation levels were checked by mass spectrometry (MS) and the labeled cells were stored as frozen stocks in DMEM with 10% DMSO at −150 °C. For the experiment, the labeled cell stock was cultivated in “heavy” medium supplemented with proline (300 mg/mL) to minimize the conversion of arginine to proline [30]. Volumes of 1 × 10^7^ of cells (heavy labeled and non-labeled) were washed in PBS, harvested in PBS completed with EDTA-free complete protease inhibitor mixture (Roche) and benzonase (Merck) and disrupted in French press by one passage at 2000 psi. Total protein concentration was determined as indicated above. The recombinant GapA-Twin-Strep tagged protein was added to freshly prepared “heavy” labeled cell lysate (6 µg of GapA to 1 mg of cell lysate), incubated for 40 min at 4 °C, and the GapA protein was purified together with its bound proteins using MagStrep “type3” XT beads, as described above. The purification was performed with the same number of lysates from non-labeled cells simultaneously (control of nonspecifically bound proteins onto the purification system). The obtained eluates were mixed and examined by LC-MS/MS analysis.

### 2.9. Sample Preparation for Mass Spectrometry Analysis

The protein samples were adjusted with 25 mM ammonium bicarbonate with 10% (*w*/*v*) sodium deoxycholate monohydrate (DOC) to a final concentration 1%. Samples were next reduced with 4 mM dithiothreitol (DTT) at 60 °C for 45 min at 700 rpm, then alkylated with 16 mM iodoacetamide at RT for 30 min in darkness. The unreacted iodoacetamide was quenched with an additional 4 mM DTT at RT for 30 min. The proteins were digested with trypsin (Promega, Madison, WI, USA) overnight at 37 °C. Thereafter, 1 M hydrochloric acid was added to stop the digestion and precipitate DOC. The suspension was then mixed with an equal volume of ethyl acetate and vortexed vigorously for 1 min, centrifuged at 14,000× *g* for 5 min, and the upper organic layer was removed. The extraction was repeated three times to completely extract DOC. The aqueous phases were desalted on EmporeTM C18-SD (4 mm/1 mL) extraction cartridges (Sigma–Aldrich) and dried in a vacuum.

### 2.10. Liquid Chromatography and Mass Spectrometry Analysis

The peptides were separated by reversed-phase liquid chromatography on an Ultimate 3000 RSLCnano system and analyzed on a Q-Exactive mass spectrometer (Thermo Fisher Scientific). The sample was loaded onto a PepMap100 C18, 3 µm, 100 Å, 0.075 × 20 mm trap column with a mobile phase containing 2% acetonitrile, 98% water, and 0.05% trifluoroacetic acid. The actual peptide separation was achieved using a linear gradient (0.1% formic acid in water as phase A; 80% acetonitrile, 20% water, and 0.1% formic acid as phase B) from 4% to 34% B in 68 min and from 34% to 55% of mobile phase B in 21 min at a flow rate of 0.3 µL/min on a PepMap RSLC C18, 2 µm, 100 Å, 0.075 × 150 mm analytical column. The column temperature was kept constant at 40 °C. The full MS/Top12 data dependent acquisition was used for identifying peptides. Positive ion full scan MS spectra (*m*/*z* 350–1650) were acquired on a 1 × 10^6^ target ion population in the Orbitrap at resolution of 70,000 (at *m*/*z* 200). Precursors ions with a charge state ≥ 2 and a minimal threshold intensity of 5 × 10^4^ counts, not fragmented during the previous 30 s, were admitted for higher-energy collisional dissociation (HCD). Tandem mass spectra were acquired at a resolution of 17,500 and with other parameters set as follows: 1 × 10^5^ for the AGC target value, 100 ms for maximum the ion injection time, 1.6 *m*/*z* for the quadrupole isolation window, and 27 for normalized collision energy.

### 2.11. Data Processing and Protein Identification

Peptides and proteins were identified by searching the raw files against the protein sequences database using the Mascot v2.4.1 search engine (Matrix Science, Boston, MA, USA) within the Proteome Discoverer v2.2 software (Thermo Fisher Scientific). The *Mus musculus* database (50,943 entries, Uniprot) was appended with protein sequences of the *F. tularensis* subsp. *holarctica* strain FSC200 (1420 entries, Uniprot), *E. coli* strain K12 (4311 entries, Uniprot), and common contaminants (245 entries). The parameters applied for protein identification were as follows: trypsin was used as the enzyme, 2 missed cleavages were allowed, the carbamidomethylation of cysteine was set as a fixed modification, the oxidation of methionine was selected as a variable modification, and the mass tolerance of the precursor and fragment ions was set to 20 ppm and 0.02 Da, respectively. Arginine (^13^C_6_, ^15^N_4_) and lysine (^13^C_6_, ^15^N_2_) were set as labels in the heavy channel for peptide quantitation. A false discovery rate of 0.01, estimated by a target–decoy approach, was used for accepting the identifications on the peptide-to-spectrum match, peptide groups, and protein levels. The SILAC approach was used to reveal the candidate proteins interacting with GapA. These were expected to be detected in a single SILAC quantitation channel, contrary to background proteins, resulting in extreme-abundance ratio values. Therefore, the low-abundance resampling algorithm of the Proteome Discoverer was used to impute missing values of single-channel quantitative data on the peptide groups’ level. The GapA interacting proteins were found as significantly upregulated proteins using a global mean rank test [31] (R package MeanRankTest) with a parametric false discovery rate level set to 0.05. Only the mouse proteins quantified in all five replicates were allowed for testing. Significantly upregulated proteins were further filtered, based on the number of unique peptides ≥2, the average protein fold change ≥2, and protein intensity in the sample to refine candidate GapA interaction partners. Data visualization was carried out in Perseus v1.6.2.3 software (Max Planck Institute, Martinsried, Germany) [32]; the processed data are part of Appendix A.

### 2.12. Solid-Phase Ligand-Binding Assay

This assay was used to confirm the ability of GapA to bind to S100A6 protein and was conducted as described previously [25]. Briefly, the recombinant mouse S100A6 (MyBioSource, San Diego, CA, USA) (0.5 µg/m) was coated onto a 96-well high-binding microtiter plate overnight, then incubated with purified recombinant GapA (0.25–2.5 µg/mL) (preparation and purification described in [25]). The amount of GapA bound to the proteins was determined spectrophotometrically (450 nm) in an enzyme-linked immunosorbent assay, using anti-GapA antibody (Apronex).

### 2.13. Immunoprecipitation for Western Blotting

#### 2.13.1. Cloning

To validate selected GapA interaction partners identified by SILAC-Pull-Down, the *gapA* gene was amplified from *F. tularensis* FSC200 genomic DNA (forward primer: 5′-GGACTCAGATCTCGAGAGTTGCAATTAATGGTTTCGGTAG-3′, reverse primer: 5′-CCGCGGTACCGTCGACTTATAGAGCTCCGAAGTACTCTAC-3′). The gel-purified PCR products (Qiagen) were inserted into pEGFP-C2 expression vector (Clontech–Takara Holding, Kyoto, Japan) between XhoI and SalI restriction sites using In-Fusion HD Cloning Kit (Takara), resulting in the plasmid encoding GapA fused with the green fluorescent protein (GFP) at its N-terminus (pEGFP-C2::*gapA*). The plasmid construct was verified by DNA sequencing.

#### 2.13.2. Transfection and Immunoprecipitation

HEK293T (ATCC) cells were seeded at density of 2 × 10^6^ cells/dish in 10 cm dishes and transfected the following day with 20 µg/dish pEGFP-C2::*gapA* or empty vector pEGFP-C2 (control) using a Calcium Phosphate Transfection Kit (Invitrogen), following the manufacturer’s instructions. Cell lysates were harvested 48 h after transfection and the GapA-GFP protein was immunoprecipitated using GFP-Trap Magnetic Agarose beads (Chromotek, Planegg, Germany) according to the manufacturer’s instructions. Briefly, cells were lysed in lysis buffer (10 mM Tris/Cl, pH 7.5, 150 mM NaCl, 0.5 mM EDTA, 0.5% Nonidet P40, protease and phosphatase inhibitors) for 30 min on ice and clarified by centrifugation at 17,000× *g* for 10 min (4 °C). The supernatant was further diluted with dilution buffer (10 mM Tris/Cl, pH 7.5, 150 mM NaCl, 0.5 mM EDTA, protease and phosphatase inhibitors) and incubated with equilibrated beads for 1 h at 4 °C. After three washes in dilution buffer, the protein was eluted with 2× SDS-sample buffer (5 min, 95 °C) and the precleared supernatant was further analyzed by immunoblot analysis, as described above.

### 2.14. Statistical Analysis

Unless otherwise stated, each experiment was independently repeated at least three times, and the assay was performed in triplicate for each time interval and strain in an experiment. The Prism 6 program (GraphPad, San Diego, CA, USA) or Excel were used for statistical analysis. The data are presented as means with standard error of the mean (± SEM) and analyzed for significance using one-way analysis of variance (ANOVA, San Francisco, CA, USA) with a recommended multiple comparison posttest. A *p*-value ˂ 0.05 was considered to be statistically significant.

## 3. Results and Discussion

### 3.1. Distribution of F. tularensis GapA in Infected Macrophages by TEM

In our recently published study, we showed that GapA protein contributes to the pathogenesis of tularemia [25]. Deleting *gapA* from the bacterial chromosome resulted in a viable mutant strain with an attenuated phenotype in both in vivo and in vitro models. Disruption of the glycolytic pathway may contribute to this characteristic, but *Francisella* is well known for its preferential utilization of specific amino acids for energy production when inside the host cell [33,34]. We were able to demonstrate the secretion and surface exposition of GapA in *F. tularensis* cultivated extracellularly. This naturally raised the question concerning the GapA localization of *F. tularensis* residing inside the host cell. To explore this, the BMDMs were infected with wt strain for 2, 12, and 24 h and processed for TEM examination. Samples with different durations of infection displayed distinctive abundance of *F. tularensis* bacteria in the cytoplasm of the BMDMs upon observation in TEM. After 2 h of infection, a cross-section of the host cells revealed the presence of approximately 1 to 5 bacteria per cell cross-section, either individually or in isolated groups (Appendix A). This number increased dramatically 12 h p.i., at which time the cytoplasm of the host cells was packed with bacteria, usually in one or two large clusters (Appendix A). Host cells in the variant 24 h p.i. were in poor condition, due to an overabundance of bacteria; hence, an analysis of this variant was not possible. Thin sections of infected cells were subsequently immunolabeled with anti-GapA antibody to reveal localization of the protein in the bacterial cells and, eventually, in the host cells upon secretion. Observation of the immunogold-labeled sections in TEM enabled us to detect the GapA of *F. tularensis* localized inside the BMDMs, not only in bacterial cell interior (Figure 1A) but also in the peripheral part of the bacteria (Figure 1B), inside the host cell cytoplasm (Figure 1C), and even in the host cell nucleus (Figure 1D). Interestingly, whereas at 2 h p.i. the distribution of GapA on the cross-section of bacterial cells was almost even, significantly more protein could be detected in the peripheral region of the bacteria at 12 h p.i. (Figure 2 and Figure 3).

The surface localization and/or release of GAPDH into the culture medium have been displayed in a number of predominantly extracellular bacteria (e.g., *Streptococcus* ssp., *Staphylococcus aureus*, and *Mycoplasma pneumonia*) [6]. Due to its proven ability to bind to various host serum and extracellular matrix proteins, GAPDH is presumed to be utilized for adhesion and invasion in these predominantly extracellular pathogens. In intracellular pathogenic bacteria, however, GAPDH might fulfill additional functions, due to their intracellular life stage, but studies with such pathogens are so far few in number. In *Mycobacterium tuberculosis*, GAPDH was identified as a surface receptor for the epithelial growth factor [35] and, in later studies, for human transferrin and lactoferrin to ensure a sufficient iron supply [8,9]. In another published study, the authors performed a fluorescence microscopic analysis of cells infected with *Listeria monocytogenes* [13]. From their observations, they deduced that this bacterium secreted GAPDH into the phagosome, where it exerted ADP-ribosylating activity on Rab5a protein, thereby inhibiting its functions. In this way *Listeria* abrogates phagosome–endosome fusion and escapes into the host cytosol for further replication. Here, we provide the first direct in situ evidence for the surface localization and secretion of a GAPDH homologue in a bacterium localized inside the host cell. Moreover, the peripheral distribution increases with infection time. This indicates that *F. tularensis* deliberately directs this protein to a location where it performs other functions, possibly relating to intracellular survival. Thus, these observations strongly support the hypothesis of direct GapA implication in host cellular processes.

The detected nuclear localization is another interesting observation. Mammalian GAPDH nuclear translocation—usually in response to cell stress—has been reported from numerous studies [1,4]. Within the nucleus, GAPDH can participate in DNA repair, as well as in the regulation of gene transcription and cell cycle. The multitasking proteins are usually highly conserved proteins in both eukaryotes and prokaryotes [36]. By having some sequence areas identical, proteins from different organisms might be able to perform similar functions if they are placed in the same environment. The *F. tularensis* GapA reveals about 41% homology with human GAPDH, based on protein BLAST protein sequence alignment (https://blast.ncbi.nlm.nih.gov/Blast.cgi, accessed on 27 January 2020). This raises the question of whether the bacterial homologue might imitate some of the functions of the host counterpart. Nevertheless, the localization of bacterial GAPDH homologues inside the host cell nucleus has not yet been described anywhere, so the real function of this phenomenon remains obscure.

### 3.2. Identification and Validation of Potential GapA Protein Interaction Partners

The additional activities of a multitasking protein often depend on its ability to associate with other proteins. To obtain further insights into the possible function of bacterial GapA protein secreted into the host cell milieu, we next investigated potential host interaction partners. For this purpose, we employed a quantitative proteomic approach based on SILAC metabolic labeling in combination with pull-down, followed by mass spectrometry analysis. A similar approach was used successfully in our previous study for the identification of GapA bacterial interaction partners [26] and for the analysis of *Salmonella* secretome and host binding partners [37]. The main advantage of this combined technique is the confident discrimination of specific and nonspecific interactions, which is the main challenge of classical protein–protein analysis methods based on co-purification itself. In our strategy (Figure 4) the lysates prepared from metabolic-labeled macrophages (J774.2 cell line) were incubated with recombinant-purified GapA protein fused with a Twin-Strep tag. Protein complexes (GapA + effector) were purified using the StrepTactin. Control non-labeled (“light”) cell lysates were manipulated in the same way. The eluates were then pooled and subjected to mass spectrometry. The specific GapA interactors were highly enriched in heavy-labeled amino acids and consequently showed a high heavy:light SILAC ratio. Non-specifically interacting proteins were characterized by equal peak intensities for both heavy and light forms.

Based on the evaluation criteria described in the Materials and Methods section, seven potential GapA binding partners were identified in this screen (Table 1): three tubulin subunits, one ribosomal protein, one protein of translation initiation complex (eIF4H), and two multifunctional proteins (DDX3X and S100A6) involved in multiple cellular processes. The interaction of eukaryotic GAPDH with tubulin is generally known and has been described in a number of studies (reviewed in [1]). The exact role of GAPDH in this interaction remains unknown. One study indicated that protein might be involved through an interaction with tubulin in the modulation of membrane trafficking and membrane fusion [38].

The interaction with DEAD-box helicase DDX3X and translation initiation factor eIF4H has not yet been described. As a cofactor, eIF4H enhances the helicase activity of eIF4A [39]. The activity of eIF4A is further stimulated when present together with eIF4E and eIF4G in the multimeric complex eIF4F. This complex mediates the recruitment of ribosomes to mRNA, thereby promoting eukaryotic translation initiation. In a study by Fiume et al. [40], the p65 subunit of NFκB was able to activate transcription of the eIF4H gene that contributes to the increased rate of global protein synthesis associated with the activation of this pathway. The helicase DDX3X, meanwhile, seems not to be essential for general translation but is needed only by some specific transcripts. It can bind to the RNA either directly or in association with components of the eIF4F complex [41] and it is possible to repress or facilitate the translation process. In addition, DDX3X enhances transcription by cooperating with various transcription factors (e.g., SP1) or influencing promoters directly (e.g., E-cadherin and IFN-β promoters). Further roles in pre-mRNA splicing, RNA export, and microRNA expression have also been described. DDX3X has been implicated in a number of other cellular processes, including cellular stress response, cell cycle, and cancer progression (reviewed in [42]). The important role of DDX3X in anti-infection innate immune response has also been demonstrated in several studies [43,44,45]. In infected cells, the helicase was found to regulate the expression of proinflammatory cytokines either by mediating translational control of various signaling pathways [43] or by modulating phosphorylation of the NF-kB signaling pathway components p65 and IKK-β [45].

To further validate the interaction of GapA with eIF4H and DDX3X, we cloned the *gapA* gene into a mammalian expression vector, pEGFP-C2. The vector was then transfected into the HEK293T cells and the expression and purification of GapA with N-terminally fused GFP was verified by immunoblot (Appendix A). In the next step, we tried to detect both the potential interactors in eluates of GapA-GFP-expressing cell line by western blot analysis using specific antibodies against eIF4H and DDX3X. However, we were able to detect only the DDX3X protein in the eluates of GapA-GFP-expressing cells. As it was absent in both negative controls (non-transfected cells and cells transfected with empty vector expressing GFP) (Figure 5), the DDX3X might be considered as an interacting partner of bacterial GapA protein. The eIF4H protein failed to be detected in the eluates. Moreover, it could be detected in the flow-through fraction obtained during the purification procedure, indicating that this protein might not directly interact with the GapA (not shown). On the other hand, the identification of eIF4H in our interaction screen could be a consequence of the GapA–DDX3X interaction that is known to associate with components of the eIF4F complex, as mentioned previously.

The validation of another potential interactor detected in our screen, the S100A6 protein, could not be performed by this approach, as this protein is not expressed in the HEK293T cell line. (https://www.proteinatlas.org/ENSG00000197956-S100A6/cell+line, accessed on 10 December 2020). We decided to confirm the S100A6–GapA interaction using solid-phase ligand-binding assay (Figure 6). In this assay, the purified GapA protein was shown to be able to bind to recombinant S100A6. GapA’s interaction with plasminogen was established in our previously published study [25] and was used as positive control here. The interaction of S100A6 with mammal GAPDH has already been observed [46], but the importance and function of this association remains obscure. S100A6, also known as calcyclin, is a small, 10.5 kDa calcium-binding protein belonging to the S100 protein family. It is localized mainly in the cytoplasm and has been shown to bind to many proteins with a prevalence of cytoskeletal proteins, tetratricopeptide repeat (TPR) domain-containing proteins, or transcription factors of the p53 family. Thus, it is suggested to be implicated in various cellular processes, especially in the regulation of cell proliferation, the differentiation and apoptosis, cytoskeletal dynamics, and tumorigenesis [47,48]. Transcription of the *S100a6* gene is regulated by several factors. While NF-κB activates the S100A6 promoter, p53 interferes with NF-κB, leading to the suppression of transcription. The overexpression of S100A6 is associated with increased activation of p38 and/or ERK kinases, indicating its involvement in pro-survival signaling [48]. Only a few studies, to date, have focused on a potential role for S100A6 in cellular processes affected by infection. Increased *S100a6* gene transcription has been demonstrated in macrophages infected with *Haemophilus parasuis* [49] or a highly pathogenic porcine reproductive and respiratory syndrome virus [50]. Recently, S100A6 has been found to interact with a surface antigen of *Toxoplasma gondii* and decreased S100A6 expression resulted in disturbed parasite invasion. This interaction enables the parasite to regulate host cytoskeleton organization and also TNF-α expression through NF-κB signaling [51].

Based on this knowledge, we decided to examine more closely the potential role of S100A6 in cells infected with *F. tularensis*. For this purpose, we applied CRISPR/cas9 technology and successfully prepared J774.1 macrophage-like cells with a knocked out *S100A6* gene (see Appendix A). The cells were then infected with a fully virulent wt strain. The bacterial invasion and proliferation rates were established by a CFU enumeration method and compared with those in infected cells having the gene preserved. Unexpectedly, there were no differences in the invasion and proliferation of bacteria in the two tested cell lines (Appendix A—Appendix A). Moreover, Western blot analysis of the phosphorylation status of the aforementioned MAP kinases (p38 and ERK) and the caspase-3/-7 activity determination by the luminescent Caspase-Glo 3/7 Assay (Promega) were not affected by eliminating the *S100a6* gene (data not shown). Thus, our preliminary functional screening failed to prove a significant role in *F. tularensis* infection. This does not, however, rule out the possibility that the S100A6 might play a minor role in some of the cellular processes affected by the intracellular pathogen and that its absence is balanced out by other factors.

### 3.3. Effect of wt and ∆gapA on Gene Expression for Selected Cytokines

A previously performed phenotype characterization of ∆*gapA* revealed significantly slower replication inside the host cell and decreased cytopathogenic effects, whereas its entry and phagosome escape remained undisturbed [25]. We wanted, therefore, to explore more closely the difference in basic host cellular processes induced by ∆*gapA* compared to the wt strain. We focused on the expression profile of selected cytokines of both pro- and anti-inflammatory character in BMDMs infected for 8 h with wt or ∆*gapA* strains. Using q-RT-PCR, we observed that the amounts of mRNA transcripts for proinflammatory tumor necrosis factor α (TNF-α), interleukin (IL)-1β, IL-6, and mediator inducible nitric oxide synthase (iNOS) were nearly the same in BMDMs infected with wt as in the control noninfected cells (Figure 7). On the other hand, the ∆*gapA* mutant strain evidently induced transcription of those cytokines. The transcripts of arginase and IL-12b were under the detection limit in all three groups (control—noninfected cells, wt infected cells, and ∆*gapA* infected cells). Surprisingly, ∆*gapA* also significantly increased the expression of IL-10 mRNA in contrast to wt. Based on the expression profile of cytokines, it seems that the murine macrophages infected with attenuated ∆*gapA* tend to be alternatively activated to M2b subtype. The M2b-polarized macrophages are generally characterized by a high IL-10/IL-12 ratio together with the expression of TNF-α, IL-6, and iNOS. Although they regulate the immune and inflammatory response, they can promote the persistence of bacterial infection. Thus, this might be related to the persistence of the ∆*gapA* mutant in the organs of infected mice that we observed in our previously published study [25]. The alternative activation of macrophages infected with *F. tularensis* LVS has been demonstrated by Shireay et al. [52]. Those authors assumed that LVS uses this mechanism to evade the host immune response.

In our experimental conditions, we were able to determine only a slight, statistically insignificant elevation of interferon (IFN) β1 transcripts (Figure 7) in BMDMs infected with both indicated strains. IFN-β is the first type I IFN produced during infection and is usually induced by bacterial DNA. Its role for the host cell can be either detrimental or protective, depending on the pathogen species. Whereas Fabrik et al. observed only low expression of this cytokine in FSC200-infected, bone-marrow-derived dendritic cells [53], upregulated transcription of IFN-β was found in phagocytes infected with *F. novicida* [54] or *F. tularensis* LVS [53,55]. *F. novicida*, as well as mutants with hypercytotoxic phenotypes, have been shown to undergo enhanced intramacrophage lysis [21,24,56]. The bacterial dsDNA released in this way is then detected by specific sensors, followed by the secretion of type I IFN required for inflammasome activation and a full proinflammatory response, including the maturation and secretion of IL-1β. The Western blot analysis of IL-1β from BMDMs infected with wt or the ∆*gapA* strain revealed that at 4 h p.i. both strains induced expression of pro-IL-1β, while at 24 h p.i. the precursor could be detected only in BMDMs infected with the ∆*gapA* mutant strain (Figure 8). The mature form of IL-1β (17-kDa), however, was not obvious under any conditions, indicating that both the strains seem not to activate the inflammasome, which serves to trigger maturation and secretion of IL-1β. Strengthening this observation, IL-1β secretion could not be detected using ELISA at 4 or 24 h p.i.; in addition, no activation of the inflammatory caspase 1 was obvious using the Caspase-Glo1 Inflammasome Assay from Promega (data not shown). To summarize this, the ability to activate the inflammasome is well established for non-pathogenic *F. novicida* and attenuated *F. holarctica* LVS, but *F. tularensis* subsp. *tularensis* infection elicits almost no inflammasome activation [21,55,56]. Our observation leads us to assume that the fully virulent *F. holarctica* FSC200 did not induce inflammasome and the GapA protein plays no role in it.

### 3.4. Effect of wt and ∆gapA on Intracellular Signaling

The transcription of genes encoding cytokines and other immune mediators is controlled by transcription factors whose activities are regulated by numerous signaling pathways, especially the NF-κB and MAPK pathways, the last named of which includes ERK, p38, and JNK kinases. The main characteristic of intracellular pathogens is their ability to disrupt these pathways to subvert the immune response [57]. In in vitro infection models, *F. tularensis* has been shown to modulate multiple signaling pathways (ERK, JNK, and p38) through temporal changes in the phosphorylated states [53,55,58,59]. To date, several *F. tularensis* factors—intracellular growth locus C (IglC) [60,61], antioxidant enzymes catalase (KatG) [62], superoxide dismutases SodB and SodC [63], RipA [64], and OmpA-like protein [65]—have been demonstrated to interfere with these signaling pathways. To determine whether the observed differences in cytokine gene transcription between wt- or ∆*gapA*-infected BMDMs might be due to differences in the induction of MAPKs, we next followed the phosphorylation status of the main kinases within the first hour of infection. A significant difference was observed in ERK 1/2 activation. The wt induced only transient activation of ERKs, which peaked at 30 min and was followed by rapid decline at 60 min p.i. On the other hand, sustained ERKs activation at 30 and 60 min p.i. was evident in cells infected with ∆*gapA* (Figure 9). The activation profile of SAPK/JNK was similar, but no changes could be seen between the two analyzed strains (Appendix A). For p38, the findings were very inconclusive, due to the extremely modest levels of phosphorylation. Our data demonstrate that the wt strain interferes with the activation of MAPK signaling pathways in the early stage of infection, whereas the ∆*gapA* mutant is not able to dampen the activation of ERKs. To investigate the NF-κB activation, we monitored the nuclear translocation of p65 subunit 1 h p.i. by immunofluorescence microscopy (Figure 10 and Appendix A). In noninfected control cells, the p65 subunit was obvious only in the cytoplasm 1 h p.i. In cells infected with wt or ∆*gapA*, the p65 subunit colocalized predominantly with the cell nucleus, indicating that both strains induce NFκB signaling. The findings were further corroborated by monitoring the status of IκBα (see Appendix A—Appendix A). Taken together, the different ability of the wt and ∆*gapA* strains to affect the ERK/MAPK signaling pathway seems predominantly responsible for the distinct transcription of genes encoding numerous cytokines and, thereby, also for the impaired proliferation of the mutant strain.

The intracellular pathogens have devised distinct mechanisms to interfere with the cell signaling on multiple levels [66]. Many such bacteria have been found to produce specific effector proteins that frequently target kinase-signaling cascades (e.g., Osp proteins of *Shigella* spp. and LegK1 in *Legionella pneumophila*). Here, we show that the bacterial GAPDH homologue might also contribute to the host signaling distortion. Whether this is the direct effect of the protein itself remains questionable, as the deletion of the whole *gapA* gene led inevitably to the disruption of bacterial glucose metabolism, resulting in a pleiotropic phenotype of the ∆*gapA* strain. Further studies should be performed that will allow for differentiating between the enzymatic and non-enzymatic functions of this protein in relation to host cell processes. Although no other study has yet addressed the potential effect of a bacterial GAPDH homologue on signaling pathways or cytokine expression, the human GAPDH is known to impact on the production of proinflammatory cytokines in response to lipopolysaccharide. According to the metabolic state, GAPDH binds to TNF-α mRNA and thus blocks its translation. Upon lipopolysaccharide stimulation, the protein is malonylated and dissociates from mRNA to promote TNF-α production [67,68]. Recently, GAPDH of the parasite *Leishmania* has been found to inhibit TNF-α expression in macrophages as well [69]. The detection of *F. tularensis* GapA inside the host cell nucleus presented in this study leads us to assume that it binds to nucleic acids while possibly influencing gene expression.

## 4. Conclusions

The data presented here follow up on our two previously published studies, the results of which pointed to potential additional functions of *F. tularensis* GapA beyond its main role in glycolysis [25,26]. Here, we focused on the potential role of GapA in host–pathogen interaction. First, using immunoelectron microscopy, we were able to detect the GapA protein not only in the peripheral part of the bacterial cell but also inside the host cell cytoplasm and nucleus. This observation points to some role of GapA during the intracellular life cycle of this pathogen. To obtain a better understanding of this, we next performed a quantitative proteomic screen to identify potential intracellular host binding partners. None of the identified hits have yet been described for a bacterial GAPDH homologue. By contrast, the interaction of eukaryotic GAPDH with tubulin or S100A6 protein has been reported previously, and this might be a consequence of some similarities in protein sequences, as GAPDH is known to be highly conserved. The interaction of GapA with DDX3X and S100A6 was confirmed by another technique. Both these proteins are multifunctional and play various roles in numerous cellular processes. It is thus impossible to formulate more exact hypotheses about the role of GapA in host–pathogen interactions. Unfortunately, no essential role of S100A6 for the course of tularemia infection in BMDMs could be estimated from our preliminary analysis. A comprehensive understanding of the importance of GapA binding to this and other identified host proteins remains a subject of further research in our laboratory.

Further, the basic cellular events responsible for the disturbed proliferation of *F. tularensis* with deleted *gapA* gene inside macrophages were explored more closely. We found that the ∆*gapA* triggered a transcription of genes encoding several cytokines whose mRNA remained at the basal level in cells infected with a fully virulent wt strain. On the other hand, it seems that both the wt and ∆*gapA* strains do not induce the activation of inflammasome, but the screen of selected signaling pathways revealed differences in the activation of the ERK/MAPK signaling pathway between the two tested strains. Nevertheless, because the pleiotropic effect of the GapA protein cannot be ruled out, this observation can be a consequence either of GapA multifunctionality or just of the disturbed glycolytic metabolism of the bacterium.

This study provides new and previously unpublished data in several areas regarding the GAPDH homologue encoded by an intracellular pathogenic bacterium which are directly related to the host–pathogen interaction. Moreover, the data obtained from the analysis of cells infected with a fully virulent strain of subsp. *holarctica* are unique. The majority of related studies have generally been performed with an attenuated LVS strain or even nonpathogenic *F. novicida*, so their results might not reflect the real mechanisms involved in the tularemia pathogenesis. Any new findings contribute toward the better understanding of pathogenic mechanisms that is essential for the identification of new prophylactic and therapeutic approaches against tularemia. In addition, this study provided us with several stimuli for future analyses. For example, the identification of sequence domains or motifs responsible for the surface exposition and secretion of GapA would enable the construction of a mutant strain with preserved metabolism but disturbed alternative roles of GapA that result from its extracellular localization. A distinction between the metabolic and nonmetabolic functions of GapA could thus be possible. Furthermore, several observations point to the potential of GapA binding to host nucleic acids. In this way, the bacterium could directly manipulate the transcription of critical genes.

## Figures and Tables

**Figure 1 cells-12-00607-f001:**
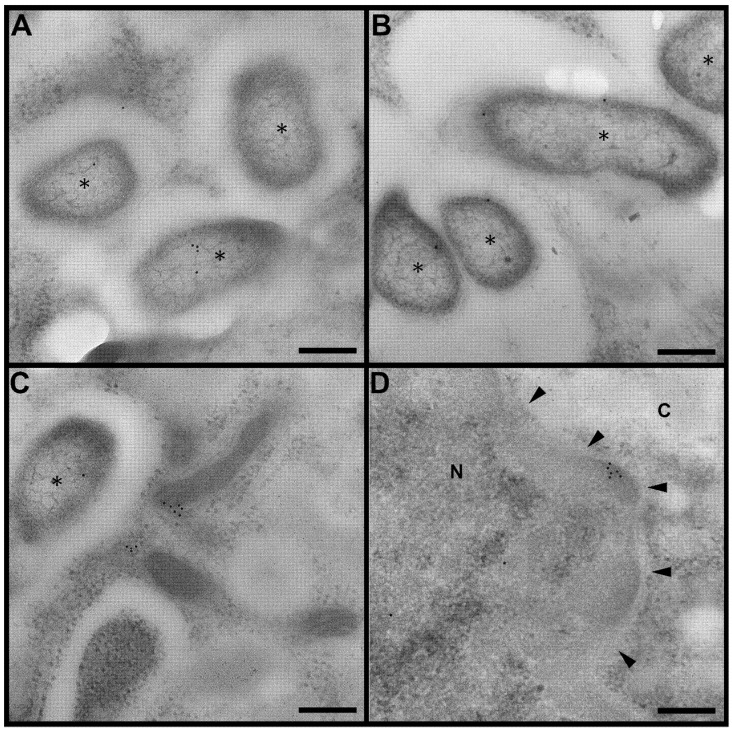
Immunogold labeling of GapA protein in infected cells. GapA is either evenly distributed inside bacteria (**A**) or predominantly localized on bacterial periphery (**B**). Additionally, GapA was detected in the cytoplasm (C) and in the nucleus (**D**) of host cells. All images display host cell 12 h p.i. Bacteria marked with asterisk (*). Images (**A**–**C**) display area in cytoplasm, image D displays nucleus (N) and cytoplasm (C) with arrowheads pointing to nuclear membrane. Scalebar = 200 nm, gold nanoparticles = 12 nm.

**Figure 2 cells-12-00607-f002:**
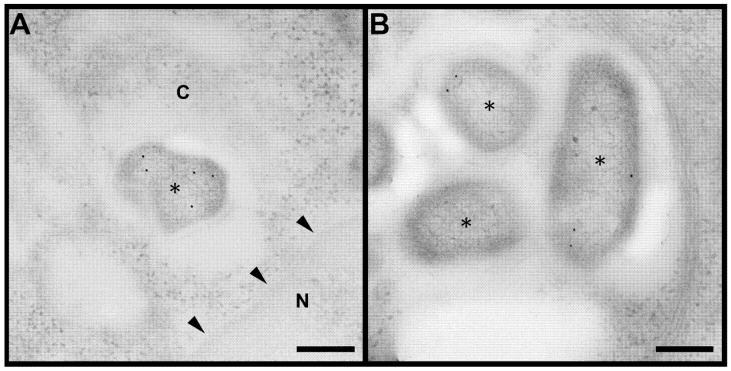
Immunoelectron microscopy analysis of GapA localization inside bacteria in infected host cell. The protein distribution was usually uniform 2 h p.i. (**A**), but apparently changed to predominantly peripheral 12 h p.i. (**B**). Bacteria marked with asterisk (*). Image A displays nucleus (N) and cytoplasm (C) with arrowheads pointing to nuclear membrane. Image B displays area in cytoplasm. Scalebar = 200 nm, gold nanoparticles = 12 nm.

**Figure 3 cells-12-00607-f003:**
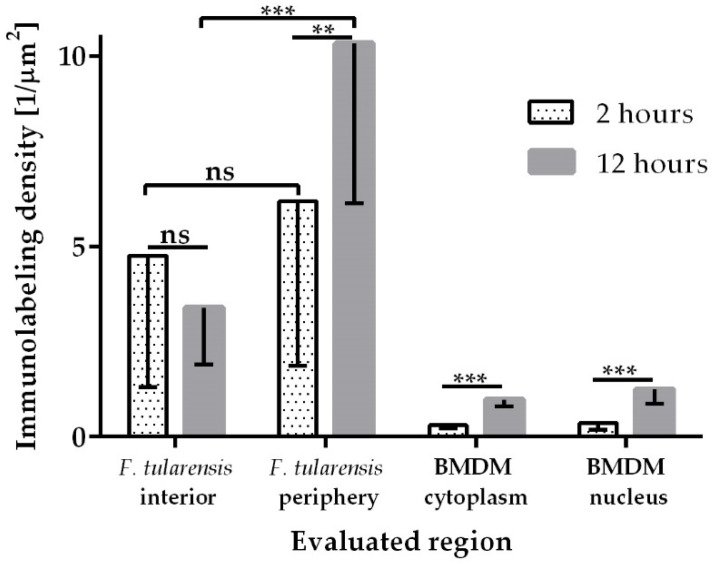
Comparison of immunogold-labeling density in specified compartments of bacterial cells and host cells. The density was calculated as a ratio between the number of gold nanoparticles in a specified region and its area in µm^2^. Statistical significance was determined by *t*-test and ANOVA (** *p* < 0.01; *** *p* < 0.001, ns = not significant *p* > 0.05). Error bars indicate the SD of values obtained from four evaluated regions of 18 randomly selected host cells per each variant.

**Figure 4 cells-12-00607-f004:**
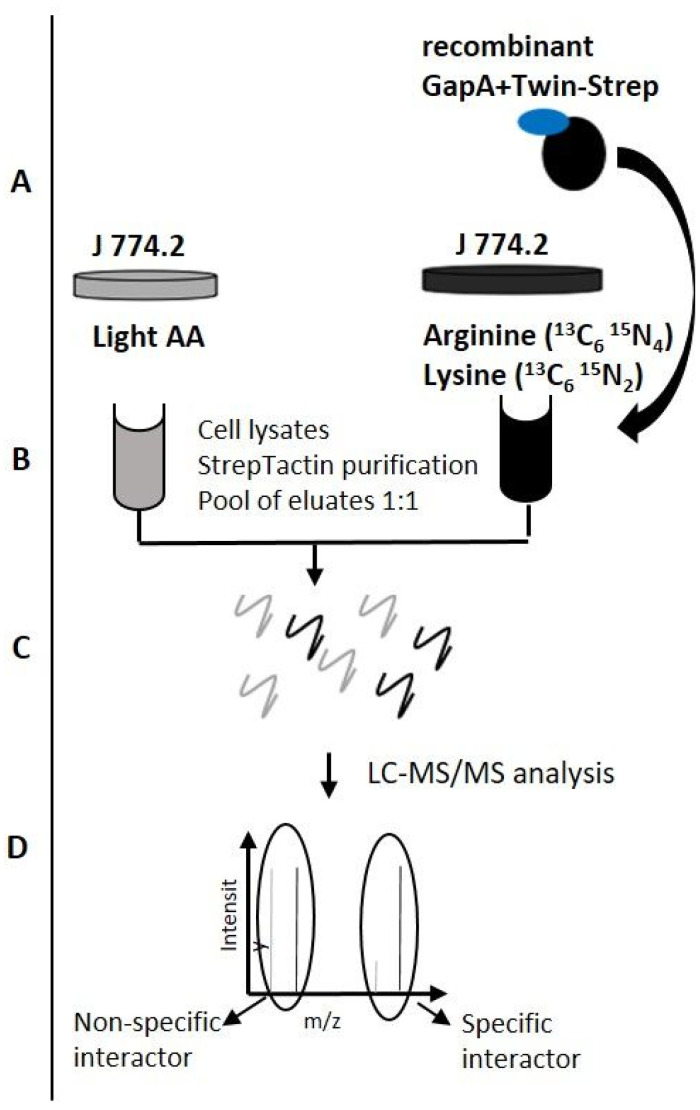
Schematic overview of the interaction partner screen. Cells were grown in medium with light or heavy amino acids (arginine ^13^C_6_ ^15^N_4_; lysine ^13^C_6_ ^15^N_2_) and lysed (**A**). Heavy-labeled cell lysate was incubated with purified Twin-Strep tagged GapA and these “heavy” lysates, as well as control “light” lysates, were subjected to purification using StrepTactin (**B**). The eluates were pooled, digested with trypsin (**C**), then analyzed by LC-MS/MS and evaluated (**D**).

**Figure 5 cells-12-00607-f005:**
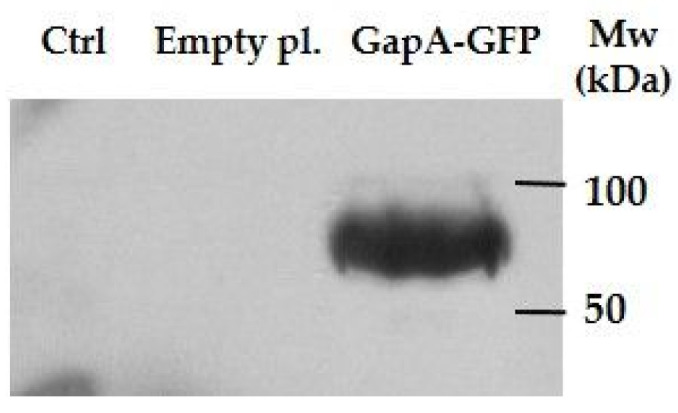
Validation of DDX3X interaction with GapA. HEK293T lysates transfected with empty plasmid pEGFP-C2 (empty pl.), pEGFP-C2::*gapA* (GapA-GFP), or non-transfected (Ctrl) were subjected to immunoprecipitation and eluates were analyzed by Western blot probing for DDX3X.

**Figure 6 cells-12-00607-f006:**
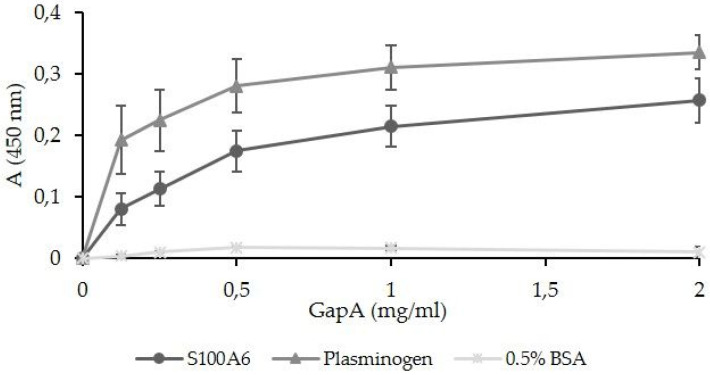
Solid-phase binding assay used for the validation of *F. tularensis* GapA binding ability to S100A6. The recombinant mouse S100A6 (MyBioSource) plasminogen for positive control and bovine serum albumin as negative control were coated on 96-well microtiter plates and reacted with different concentrations of purified GapA. Data are presented as means ± SD from three independent experiments.

**Figure 7 cells-12-00607-f007:**
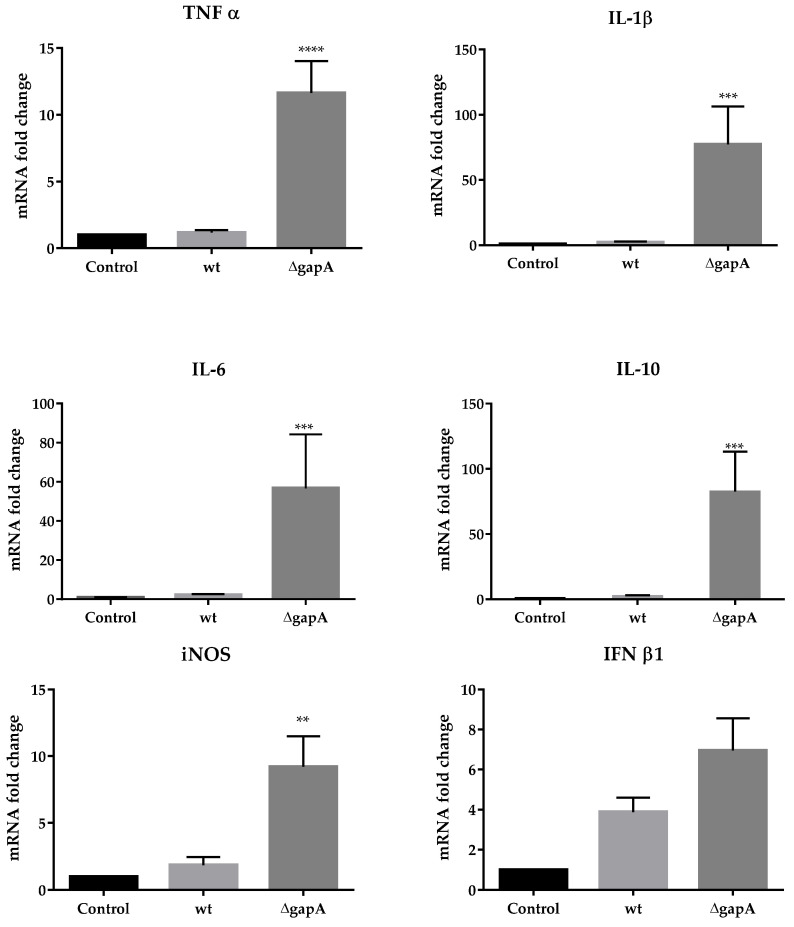
Real-time qRT-PCR quantification of selected cytokines in BMDMs infected with wt or ∆*gapA* strains for 8 h. Results were compared to uninfected cells (control). Data are presented as a fold change in expression relative to control cells (set at 1). Data are shown as mean ± SEM. ** *p* ˂ 0.01, *** *p* ˂ 0.001, **** *p* ˂ 0.0001 (two-way analysis of variance with Tukey’s posttest).

**Figure 8 cells-12-00607-f008:**
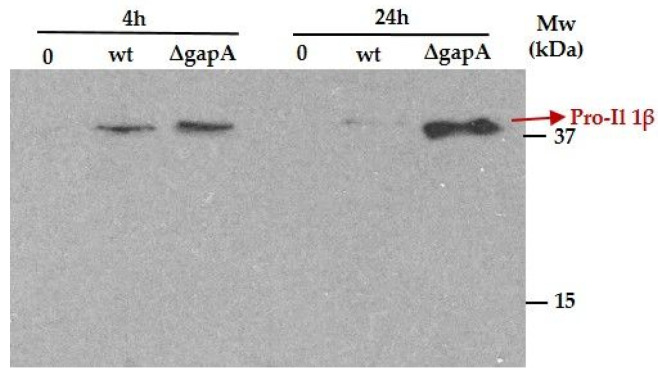
Western blot analysis for visualization of pro-IL-1β form in cell lysates from BMDMs that were noninfected (0), infected with wt strain (wt), and infected with ∆*gapA* at 4 and 24 h p.i.

**Figure 9 cells-12-00607-f009:**
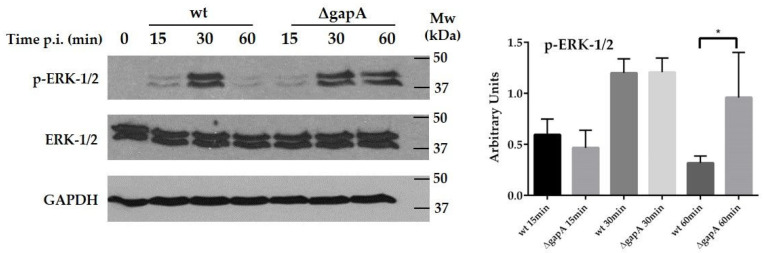
Western blot analysis of ERK 1/2 MAPK kinase in BMDMs infected with wt or ∆*gapA* at MOI 50 for 15, 30, and 60 min. Representative Western blots are shown. Bands were quantified using ImageJ (right panel, n = 4 blots). Data shown as mean ± SEM were analyzed using one-way ANOVA (* = *p* ˂ 0.05).

**Figure 10 cells-12-00607-f010:**
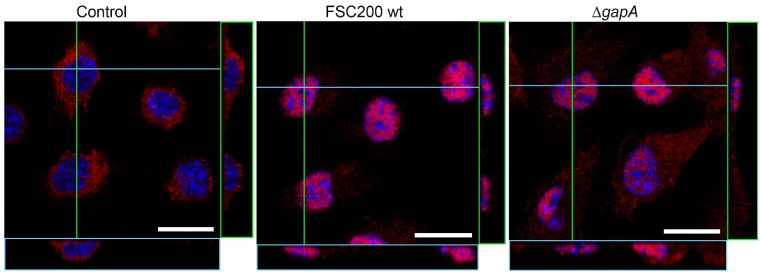
Confocal microscopy of BMDM cells (uninfected—control, or infected either with Δ*gapA* or wt) stained with Hoechst 33,342 (blue; nuclei) and NF-κB p65 (red). Each image is one confocal plane with *z*-axis projections (for blue and green lines) showing the distribution of NF-κB in the cell, especially the differences of cytoplasm versus nucleus localization. Bar in each image represents 10 μm.

**Table 1 cells-12-00607-t001:** Candidate GapA interaction partners identified by SILAC–Pull-Down–MS/MS strategy.

Accession Number (Uniprot)	Gene Name	Protein Name	Function	Subcellular Location
P99024	*Tubb5*	Tubulin beta-5 chain	Microtubule constituents	Cytoplasm, cytoskeleton
P68372	*Tubb4b*	Tubulin beta-4B chain		
P05213	*Tuba1b*	Tubulin alpha-1B chain		
Q91V55	*Rsp5*	40S ribosomal protein S5	Ribosome constituent	Cytosolic small ribosomal subunit
Q62167	*Ddx3x*	ATP-dependent RNA helicase DDX3X	Multifunctional	Nucleus, cytoplasm
Q9WUK2	*Eif4h*	Eukaryotic translation initiation factor 4H (eIF4H)	Stimulates initiation of protein synthesis	Cytoplasm, perinuclear region
P14069	*S100a6*	Protein S100-A6	Multifunctional (calcium sensor, modulator)	Cytoplasm, nucleus envelope

## Data Availability

The data are available upon reasonable request from the corresponding author.

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
