# Peer review of "Francisella tularensis Glyceraldehyde-3-Phosphate Dehydrogenase Is Relocalized during Intracellular Infection and Reveals Effect on Cytokine Gene Expression and Signaling"

_cells, 2023, doi:10.3390/cells12040607_

Round 1

Reviewer 1 Report

Thank you for sending me the research article paper “Francisella tularensis Glyceraldehyde-3-Phosphate Dehydrogenase is Secreted during Intracellular Infection and Reveals Pleiotropic Effect on Cellular Pathogenesis” for review in the Cells. In the article of Ivona et al., the author discussed the role of GAPDH in the infection of F. tularensis and pathogenesis. There are important points that should be discussed and improved.

1.         Author should re-write the title. It looks general. There is a need to include manuscript conclusions.

2.         Introduction part is too long and general. Author should write an introduction in a scientific way. There is a need to write sentences with proper words or data. General statements have no worth. Paragraphs 2nd and 3rd should be combined and written in a precise way.

3.         Author should mention how macrophage cells are characterized and selected.

4.         The Method section should be written in a precise way. There is too much generalization and useless information.

5. Author should revisit the manuscript and re-write it in a more precise and scientific way. 

Author Response

We thank the reviewer for critical reading and pragmatic comments and suggestions. We tried to make appropriate changes according to your suggestions. Answers to your comments and questions are listed below.

to point 1:

The title of the manuscript has been changed.

to point 2, 4, 5:

We tried to improve the text. However, as the manuscript has been revised by two other reviewers, we have to respect also their opinions and comments. All the changes in the text are marked.

to point 3:

The bone-marrow derived macrophages are one of the most used macrophage models in many studies. The isolation and differentiation was performed according to generally used protocol (e.g., Toda et al., STAR Protocols 2, 100246, 2021, https://doi.org/10.1016/j.xpro.2020.100246). These macrophages are fully differentiated in vitro from bone marrow stems thus being naive and unpolarized at the time of harvesting. The differentiation of stem cells to macrophages is promoted by macrophage colony stimulating factors (M-CSF). The L929 conditioned medium is usually used as an alternative source of these factors, as L929 cells secrete them into the supernatant. Over 90% of cells differentiated in this way are positive for F4/80 marker.

Reviewer 2 Report

Authors‘ findings are interesting. Several questions need to be addressed, as follows:

(1) The quality of figures 1 and 2 is poor. I suggested authors used another method to demonstrate it.

(2) Authors need performed more experiments, i.e., observation of cell growth state between wt and ∆gapA strains. 

(3) I also suggested that authors observe the numbers of bacteria in cells using the staining method. 

Author Response

We thank the reviewer for critical reading and pragmatic comments and suggestions. Answers to your comments (in bold) and questions are listed below.

  • The quality of figures 1 and 2 is poor. I suggested authors used another method to demonstrate it.

The quality of EM images in the original panel is good; visually lower contrast of the images is normal for the samples embedded in acrylic resin for subsequent immunolabeling. The contrast is also adjusted in such manner that the gold particles are well visible over the sample structure. The advantage of immunoEM method is that it allows to detect the molecules of interest in situ with high spatial resolution and unambiguous relation to cellular structures.

Prior to the immunoEM analysis, we tried to examine the GapA localization inside the infected cells also by another technique, like Western blotting or STED microscopy. However, due to the technical limitations of these procedures, the results were far from being convincing. The immunoelectron microscopy remains the unique technique that enables the direct visualization of a single protein inside the whole cell.

(2) Authors need performed more experiments, i.e., observation of cell growth state between wt and ∆gapA strains. 

(3) I also suggested that authors observe the numbers of bacteria in cells using the staining method. 

The basic phenotypic characterization of ∆gapA mutant strain and comparison to the wt strain was performed earlier and the results are shown in our previously published paper (Pavkova I, Kopeckova M, Klimentova J, Schmidt M, Sheshko V, Sobol M, Zakova J, Hozak P, Stulik J. The Multiple Localized Glyceraldehyde-3-Phosphate Dehydrogenase Contributes to the Attenuation of the Francisella tularensis dsbA Deletion Mutant. Front Cell Infect Microbiol. 2017 Dec 11;7:503. doi: 10.3389/fcimb.2017.00503. PMID: 29322032; PMCID: PMC5732180). In this study, we followed the in vitro growth, the intracellular invasion and replication, viability of infected cells and cytotoxic potential of both the strains. The performance of immunofluorescence analysis is also mentioned, which excludes its presentation in this manuscript.

Reviewer 3 Report

In the currect paper the authors try and answer the questions extending from their previous paper on F. tularensis GapA. The authors show differential localization of the GapA within the bacteria and within the host and also identify its interacting partners in the host. The paper further makes use of GapA Knockout strain to show its effect on cytokine gene expression and signalling.  The paper is well written with detailed description. 

Concerns and suggestion:

- Authors make a firm claim on secretion of GAPDH in host cytoplasm and nucleus and also claim it in the title. Although the TEM images clearly show that, but it needs to be supported with proper controls (using an isotype control or using GapA Knock out strain). How specific is GapA antibody since it is 41% homology to human GAPDH? For calculating statistical significance (fig 3) it's hard to understand if the regions selected were from single replicate or multiple replicates. Also, using an alternate method to show secretion of protein in the host. 

- Since the phospho western blots for signaling were performed in multiple replicates, it would be nice to see an intensity graph normalized to loading control next to the western blot and calculate the statistical significance rather than just visual observation. 

- Mass spec data processing - Cut off for selection of the candidates can be mentioned in detailed (fold change?). If there is any excel output file of the mass spec data which provides any additional information, then it may be uploaded as supplementary file. 

- Line 622-624 - couldn't find the data supporting this, is the data not shown?

- Line 630-631 - authors claim gapA deletion has no effect because the WT had no effect is probably not the correct way to put it. It just probably means gapA plays no role in this. 

- Line 673 - Mentioning few cells is a relative term, authors need to provide a number or a statistical value. After providing some statistics, authors can claim things (line 675 onwards)

- Fig 10 - images can be rearranged as control - WT - gapA deletion (as followed for previous figures)

- Conclusion can be shortened, some sentences seem to be a repeat from results and discussion.

- Supp fig S7 and S8, p65 (red color) in gapA deletion strain seems very intense compared to WT. It would be great if authors could comment on this. 

Author Response

We thank the reviewer for critical reading and pragmatic comments and suggestions. We tried to make appropriate changes according to your suggestions. Answers to your comments (in bold) and questions are listed below.

Concerns and suggestion:

- Authors make a firm claim on secretion of GAPDH in host cytoplasm and nucleus and also claim it in the title. Although the TEM images clearly show that, but it needs to be supported with proper controls (using an isotype control or using GapA Knock out strain). How specific is GapA antibody since it is 41% homology to human GAPDH?

We can claim that the immunolabeling with GapA antibody is specific for the bacterial protein based on labelling of uninfected cells used as a control. The labelling density (gold particles per µm2) over the cellular sections without infection was 0.31±0.09, which was comparable to the negative control (incubation with secondary antibody only): 0.20±0.13. We finally also decide to mitigate the claim on secretion in the manuscript as well as in the manuscript title, as this finding is only based on the observation of protein localization (statistically evaluated), and we can only indirectly assume about the mechanism.

The specificity of the GapA antibody has also been thoroughly tested by Western blot, immunofluorescence microscopy. Despite the homology, the antibody revealed relative high specificity in all these techniques.

For calculating statistical significance (fig 3) it's hard to understand if the regions selected were from single replicate or multiple replicates. Also, using an alternate method to show secretion of protein in the host. 

The data originate from one experiment, where two different TEM grids for each variant were incubated in parallel. Calculation of statistical significance was based on comparison of labelling density values and variance in individual cells between analysed variants.

Prior to the TEM analysis, we tried to examine the GapA localization inside the infected cells also by another technique, like Western blotting or STED microscopy. However, due to the technical limitations of these procedures, the results were far from being convincing. The immunoelectron microscopy remains the unique technique that enables the direct visualization of a single protein inside the whole cell.

Changes in the text regarding the used controls and evaluation are highlighted in yellow.

- Since the phospho western blots for signaling were performed in multiple replicates, it would be nice to see an intensity graph normalized to loading control next to the western blot and calculate the statistical significance rather than just visual observation. 

For phosphorylation of ERKs shown in the main manuscript, we added a graph with normalized intensities with marked statistical significance. The evaluation was performed using the ImageJ software. However, in our opinion, quantification of chemiluminescent western blot data, derived from film-based detection is very challenging and prone to subjective errors.

- Mass spec data processing - Cut off for selection of the candidates can be mentioned in detailed (fold change?). If there is any excel output file of the mass spec data which provides any additional information, then it may be uploaded as supplementary file. 

Thank you for your comment, we added the required data to the text (section 2. Materials and Methods – 2.11. Data Processing and Protein Identification – highlighted in yellow) and uploaded an Excel output file with MS data as supplementary file.

- Line 622-624 - couldn't find the data supporting this, is the data not shown?

In Western blot (Fig.8) only bands corresponding to the pro-IL-1β (about 37 kDa) form are visible, there is no band around the 17 kDa corresponding to the mature form of IL-1β. The data from ELISA as well as caspase-1 activation assay are not shown. Minor text editing makes this hopefully clearer (highlighted in yellow).

- Line 630-631 - authors claim gapA deletion has no effect because the WT had no effect is probably not the correct way to put it. It just probably means gapA plays no role in this. 

Thank you for your suggestion, the text was changed accordingly (highlighted in yellow).

- Line 673 - Mentioning few cells is a relative term, authors need to provide a number or a statistical value. After providing some statistics, authors can claim things (line 675 onwards)

We fully accept your comment. The main output of this analysis is the demonstration of nuclear translocation of p65 into the nucleus which reflects the activation of NFκB signaling pathway. This process could be seen in cells infected with wt and with ΔgapA, indicating, that 1 h post infection this signaling pathway is activated in response to the presence of both the F. tularensis strains. In some ΔgapA cells, the red signal was obvious also outside the nucleus, and we only wanted to comment this additional observation. The reason for this phenomenon remains questionable and further experiments based on the analysis of multiple time intervals or better time-lapse microscopy would be necessary. We thus decided to remove the relevant part from the text.

- Fig 10 - images can be rearranged as control - WT - gapA deletion (as followed for previous figures)

The images in the Figure 10 were rearranged according to your suggestion.

- Conclusion can be shortened, some sentences seem to be a repeat from results and discussion.

We modified the text in part “Conclusions”.

- Supp fig S7 and S8, p65 (red color) in gapA deletion strain seems very intense compared to WT. It would be great if authors could comment on this. 

On the graph below the images you can see, that the maximum red signal (for p65) overlaps with the maximum blue signal (nucleus) in both the cells (infected with wt and ΔgapA) indicating the p65 co-localization with the cell nucleus. In non-infected cells, the signal seems to be lower but it is evenly distributed through the cell without any peak. In cells infected with the wt strain all the red signal overlaps with the blue one, indicating that all the labeled protein is situated in the nucleus at this time point. In ΔgapA infected cells the red signal starts to increase even before the blue one; however, the maximum red signal overlaps with the blue one. This indicates, that most of the protein is located inside the nucleus but some part of the protein is still outside the nucleus. The total red signal seems to be higher in these cells and one could speculate about increased expression of this protein but this should be verified by another technique (Western blot, quantitative MS). We performed a preliminary Western blot and the p65 expression looked the same for both the strains.

Round 2

Reviewer 1 Report

Author has fulfilled the desire modifications. 

Author Response

We thank the reviewer for accepting our answers to the comments and modifications. There are no other comments to be answered.

Reviewer 2 Report

I still think another method, such as WB or Immunofluorescence methods, is required in the Tables 1 and 2.  In general, immunoEM method in combination with another semi-quantitative method is better and required in this current study. 

Author Response

Than you for your comment. As already stated, we originally intended to evaluate the GapA localization inside the host cell using the methods you suggest. However, the experiments performed by WB and superresolution light microscopy (STED) yielded inconclusive results. On the other hand, the results obtained by immunoelectron microscopy provide solid evidence for localization of the bacterial GapA in the peripheral part of the bacteria and also in the host cell upon infection. The sensitivity of the IEM approach is the highest among listed, with regard to its ability to detect single molecules in the sample. The specificity of the labelling was controlled in a proper way in uninfected samples, and the increase of labelling in bacterial periphery and in BMDM cytoplasm and even nucleus upon 12 hours of infection is highly statistically significant. This unique finding was only possible due to use of electron microscopy immunolabeling, as the amount of the bacterial protein relocated to the host cell is very probably too low to be detected by biochemical methods (impossible to achieve perfect fractionation), or light microscopy (where diffuse signal of low intensity is technically difficult to threshold over background). It is also good to realize, that the GapA protein is predominantly cytosolic protein, not a classical secreted protein. We therefore demonstrate here its multiple localization. The majority of GapA remains in association with bacteria under any conditions (in the cytosol and in the periphery) and a reliable detection of the small amounts of protein localized outside the bacterial cell is very challenging. Moreover, it is essential to distinguish the signal coming from the protein localized inside and outside the bacterial cells. It is worth considering if another technique which is much less sensitive could really give us a valid information. On the other hand, if our statement was primarily based on results obtained by WB or IF, we would certainly proceed to verify them by another method.

Please note, that during the first revision round, we decided to soften our statement concerning the protein secretion and made appropriate changes in the manuscript as well as in the manuscript title.

Reviewer 3 Report

The authors have responded to my prior comments, and I have no further comments.

Author Response

(The authors gave the same response as above.)
